# Polymer Scaffolds for Biomedical Applications in Peripheral Nerve Reconstruction

**DOI:** 10.3390/molecules26092712

**Published:** 2021-05-05

**Authors:** Meng Zhang, Ci Li, Li-Ping Zhou, Wei Pi, Pei-Xun Zhang

**Affiliations:** 1Department of Orthopedics and Trauma, Peking University People’s Hospital, Beijing 100083, China; mengzh2008@126.com (M.Z.); drlici@bjmu.edu.cn (C.L.); piwei@pku.edu.cn (W.P.); 2Key Laboratory of Trauma and Neural Regeneration, Peking University, Beijing 100083, China; 3Beijing Key Laboratory for Bioengineering and Sensing Technology, Daxing Research Institute, School of Chemistry & Biological Engineering, University of Science & Technology Beijing, Beijing 100083, China; lipingzhou@xs.ustb.edu.cn; 4National Center for Trauma Medicine, Beijing 100083, China

**Keywords:** peripheral nerve injury, nerve conduits, nerve reconstruction, polymeric scaffold, natural and synthetic polymers

## Abstract

The nervous system is a significant part of the human body, and peripheral nerve injury caused by trauma can cause various functional disorders. When the broken end defect is large and cannot be repaired by direct suture, small gap sutures of nerve conduits can effectively replace nerve transplantation and avoid the side effect of donor area disorders. There are many choices for nerve conduits, and natural materials and synthetic polymers have their advantages. Among them, the nerve scaffold should meet the requirements of good degradability, biocompatibility, promoting axon growth, supporting axon expansion and regeneration, and higher cell adhesion. Polymer biological scaffolds can change some shortcomings of raw materials by using electrospinning filling technology and surface modification technology to make them more suitable for nerve regeneration. Therefore, polymer scaffolds have a substantial prospect in the field of biomedicine in future. This paper reviews the application of nerve conduits in the field of repairing peripheral nerve injury, and we discuss the latest progress of materials and fabrication techniques of these polymer scaffolds.

## 1. Introduction

Peripheral nerve injury (PNI) is a frequently occurring disease, which means that the peripheral nerve plexus, nerve trunk, or its branches are damaged by an external force. The incidence of new peripheral nerve injury is increasing worldwide, and about one million patients need peripheral nerve surgery every year [1]. Peripheral nerve injury is usually caused by traumatic lesions such as penetrating wounds, squeezing, stretching, and also ischemia [2]. From 1% to 3% of trauma patients will have an injury involving a peripheral nerve [3,4]. Although peripheral nerves have a significant ability to regenerate after injury, long-term nerve defects after peripheral nerve injury (PNI) are a challenge to entire recovery [5]. Functional recovery of peripheral nerve injury is a slow process. In order to achieve better neural function quickly, a variety of nerve suture methods and new polymer nerve conduits have been widely proposed. The development of polymer nerve conduits to enhance nerve repair remains an important research area, and future clinical developments in the area of nerve regeneration and repair could potentially improve patient outcomes.

This paper reviews the applications of different polymer scaffold nerve conduits in peripheral nerve injury, discusses the possible reasons, advantages, and disadvantages of each material in promoting nerve functional rehabilitation, and puts forward a possible direction for developing nerve repair materials in the future.

## 2. Use of Conduits

Direct tensionless end-to-end repair is considered a significant factor in the successful recovery of sensorimotor function. When there is a long peripheral nerve defect that cannot be directly repaired without tension, autologous nerve transplantation is considered the gold standard [6,7]. However, this approach often leads to obvious donor site complications. With the development of new materials, the effect of nerve conduit repair is close to, or even better than, that of nerve graft. Besides, some new materials have absorbability and good biocompatibility. No axon escape, promotes axon growth and maximizes the amplification and crossing domination of nerve innervation, making it possible for small axons to repair large axons quickly. Research into tissue engineering scaffold materials widely use natural scaffold materials, synthetic degradable polymers, and composite scaffold scaffolds.

## 3. Natural Materials

The natural biofunctional scaffold is composed of biomaterials full of promoting elements, such as nerve growth factor, vascular endothelial growth factor, and neurotrophic factor, providing a promising strategy for the regeneration of peripheral nerve defects. Various natural, synthetic, and biological materials have been applied to fabricate artificial nerve conduits. Natural-based materials are widely available, although their histocompatibility differs from that of acellular nerve grafts. For example, chitosan can be used as an absorbable natural material when used in nerve regeneration. Extracellular matrix (ECM) components have mostly been used for nerve reconstruction. Among these natural polymers are laminin, fibronectin, and collagen. Nerve conduits made of different materials have different biophysical characteristics, but there are still challenges in promoting functional rehabilitation.

### 3.1. Chitosan

As a kind of linear polysaccharide, chitosan has many similarities with the extracellular matrix and has specific tension resistance and plasticity. However, physically guided nerve catheters may not be sufficient to promote optimal recovery [8]. After implantation, the conduit should degrade as the nerve regenerates. Nerve conduits made using biodegradable materials have been considered especially promising. Water-soluble, biocompatible and non-toxic chitosan is a natural substitute and can be used as a conduit for axonal regeneration. The design of chitosan catheters can provide an environment conducive to axon regeneration and nerve regeneration, until the vascularization of regenerated tissue, and the degradation of catheter materials [9]. Many studies have shown that chitosan catheters can be used as a suitable scaffold biomaterial to promote nerve regeneration with minimum cytotoxicity and high biodegradability [10]. Another study showed a large number of axons in the chitosan embedded nerve graft group, compared with the nerve graft group alone. The purpose of an absorbable nerve conduit is to quickly provide a tranquil environment for nerve regeneration, and then degrade without an inflammatory reaction after nerve reconstruction. For defects with long nerve stump distance, studies have shown that a chitosan catheter can induce significant motor and sensory axon regeneration in a mouse model with a 10–15 mm nerve injury [11,12]. Even in the study of Ding, neural scaffolds made of chitosan and its copolymers combined with stem cells, showed certain target muscle reinnervation in the repair of a 50 mm defect model [13].

In addition to the separate application of chitosan catheters, there are many experiments in which chitosan is combined with a variety of cells, nerve growth factors, or other implants to achieve a better effect in promoting nerve regeneration. For example, Azizi et al. used a chitosan catheter to load *α*-lipoic acid to promote sciatic nerve regeneration in rats [14]. Similarly, a chitosan catheter combined with hyaluronic acid prevents sciatic nerve scar in rats with a peripheral nerve crush injury. Artificial nerve grafts composed of a mixture of chitosan and synthetic polymers have a more remarkable ability to regenerate when combined with Sertoli cells. In some studies, a chitosan conduit and SF fibers were used as fundamental biomaterials for reconstituting SC-derived ECM, and to generate a hybrid nerve scaffold [15]. It is suggested that the transplantation into an injured 10 mm sciatic nerve of tissue-engineered nerve, constructed by chitosan nerve conduit, can increase the diameter and area of axons, improve motor function and improve the electrical conductivity of regenerated axons [16].

### 3.2. EMC and Its Derivatives

Many extracellular matrix components (EMC) and their derivatives can be made into nerve scaffolds after a series of processes, including laminin, collagen, glycosaminoglycan, elastin, and fibronectin. Remarkable progress has been made in many studies on applying these components alone, or in combination with other synthetic fibers. Besides, some natural proteins, polysaccharides can also be used to manufacture biological scaffolds, such as silk fibroin, spider silk protein, keratin, and alginate, which are similar to nerve conduits made by extracellular matrix components.

#### 3.2.1. Collagen

Collagen is the main structural protein in the human body, collagen-rich nerve bridging scaffolds and collagen-binding NGF could be useful biomaterials for the repair of severe nerve injuries [17]. Compared with autogenous nerve transplantation, bridging a 30-mm nerve defect using collagen filaments achieved a good result [18]. Although the collagen nerve guide channel has been successful in nerve repair, collagen is relatively expensive, and because of the lack of mechanical strength, challenging to deal with in the suturing process [19]. In terms of combining cytokines, a comparative experiment of artificial nerve catheters has been carried out, including collagen extract alone and autologous nerve transplantation. Cui et al. used a functional collagen nerve conduit combined with nerve cytokines to bridge a 35 mm long facial nerve gap. A miniature pig model was used to evaluate facial nerve regeneration at a longer distance than the rodent model [20]. They concluded that the collagen tubules filled with BDNF-rich collagen appeared to be at least as effective as autologous nerve grafts in compensating for the short facial nerve gap [21]. Furthermore, the rats repaired by Waitayawinyu and his team using biological scaffolds made of collagen polymers, showed better isometric muscle contractility, axon count, wet muscle weight, and axial protruding bud tissue [22].

#### 3.2.2. Laminin

In addition to collagen, laminin present in the vascular basal lamina can also act as a conduit for axonal growth [23,24]. It is expressed in the basement membranes surrounding peripheral nerves, capillaries, brain, and skeletal muscle [25]. Because it can regulate the proliferation, differentiation, myelin formation, and differentiation of Schwann cells, it is considered a potential material for nerve repair [26,27]. In addition, a laminin catheter with multi-layer design is appropriate for surgical implantation. Laminin and laminin-polycaprolactone (PCL) blend nanofibers, can mimic peripheral nerve basement membrane [28]. Improvement of sciatic nerve regeneration has been shown using laminin-binding human NGF-beta [29]. Furthermore, the improvement of axon growth was also observed in nerve guides filled with laminin gel [30]. These results prove the powerful effect of laminin in promoting nerve regeneration.

#### 3.2.3. Alginate Keratin Silk Fibroin Protein

Alginate is a kind of polysaccharide widely distributed in brown algae’s cell wall. Alginate can promote Schwann cells’ survival and growth and promote the neurite germination of embryonic dorsal root ganglion neurons [31]. Fannon et al. demonstrated that the aggregation and differentiation of mouse embryonic stem cells encapsulated in an alginate scaffold, could achieve aggregates similar to embryonic stem cells in the standard environment [32]. Keratin-derived nerve scaffolds can also effectively repair 4-mm tibial nerve defects in mice and 10–15 mm sciatic nerve defects in rats [33,34,35]. In addition to alginate and keratin, silk fibroin is also widely used in biological scaffolds, and stem cells on silk fibroin scaffolds can be induced faster, and promote axonal regeneration. These non-extracellular matrix components can be polymerized or modified naturally, to produce scaffolds that are more in line with requirements.

## 4. Synthetic Polymers

Because synthetic polymers can better simulate the physical and chemical properties of nerve tissue, the polymerization or esterification between different materials may have more advantages in tissue engineering. Therefore, it has attracted significant attention in the field of biological regeneration. Like natural polymers, synthetic nerve conduits connect nerve posts’ gaps, to guide and support nerve regeneration. They can be broadly classified into degradable and non-degradable materials. However, experiments have shown that biodegradable nerve catheters are more effective in promoting the regeneration of peripheral nerves than silicone catheters [36]. This means soluble polymer nerve conduits have become a promising research direction in modern tissue engineering.

### 4.1. Synthetic Non-Degradable Polymers

The synthesized non-degradable polymer, organosilicon, is one of the earliest nerve repair materials. When, in rats, a sciatic nerve defect of 10 mm was repaired with a silicone conduit, the unmyelinated nerve fibers could be regenerated on the 16th day, and the muscle strength of gastrocnemius muscle was partially restored on the 120th day [37]. Clinical studies have shown that when, at the midpoint, the nerve is removed to form a 10 mm gap and bridged with the silicone tube, the repair effect is still considerable after 3–5 years [38]. Because silicon conduits have achieved good results in repairing peripheral nerves, it is still meaningful to undertake repair based on single-compartment silicon derivatives, or combined with nerve growth factor. However, these synthetic non-degradable scaffolds will remain in the body after axonal regeneration, causing foreign bodies or space-occupation, and limiting their development.

### 4.2. Synthetic Degradable Polymers

The ideal synthetic degradable scaffold should be biocompatible, display a macrostructure that supports cell growth and nutrient and waste exchange, and a surface that allows for cell migration, attachment, and differentiation. This yields a fully degradable, porous, highly interconnected scaffold. Comparing degradable and non-degradable scaffolds reveals that degradable scaffolds promote vascularization, in terms of the amount and invasion depth [39].

#### 4.2.1. PLA, PGA, and Its Derivatives

The nerve conduit fabrication method can also be applied to various polymers, such as polylactic acid (PLA), polyglycolic acid (PGA), and its derivatives. PLA can be extracted from renewable resources such as potatoes and sugarcane. A variety of methods can synthesize it, including ring-opening polymerization of lactide dimer in the presence of suitable catalysts, lactic acid polycondensation, or enzymatic synthesis of lipases in the absence of metal catalysts [40,41]. Because PLA has good shaping and molding properties, it is the preferred base material for nerve regeneration pipelines [42]. PLA conduits facilitate nerve regeneration in rats with sciatic nerve injury and are considered a promising alternative for autologous nerve grafts [43,44]. Lu et al. used a polylactic acid (PLA) fiber-reinforced catheter to evaluate peripheral nerve regeneration in a 10 mm rat sciatic nerve model. The nerve bundle of the PLA group was significantly more extensive than that of other groups. The reason may be that the artificial catheter provides an essential microenvironment for peripheral nerve regeneration, especially the effect of nerve amplification. However, PLA itself does not cater to any nerve regeneration activity. Compared with PGA, PLA has stronger hydrophobicity, and its degradation by-product is lactic acid, which is easily eliminated by metabolism [45]. Polyglycolic acid is a linear aliphatic polyester that can be extracted from glycolide synthesized from sugarcane or pineapple. The ester group is responsible for stabilizing the structure of the polymer. The degradation product is the glycolic acid group, which can be excreted in urine. This polymer can be prepared by ring-opening polymerization of GA [46]. Poly (*α*-hydroxy ester) polymer, is a polymer formed by polylactic acid (PLA) and polyglycolic acid (PGA). Compared with its components, poly (*α*-hydroxy ester) polymer, when used in biological scaffolds and nano-drug delivery systems, is more easily processed and shows high biocompatibility, high mechanical strength, low toxicity, biodegradability, and non-immunogenicity. Ester bond cleavage can be reduced to polylactic acid and polyglycolic acid to continue its metabolism. PLGA is a copolymer formed by PLA and PGA, and because its surface has a more microporous structure, it can better load and slowly release growth factors and other complexes, and may be more promising.

#### 4.2.2. Polyethylene Glycol (PEG)

Polyethylene glycol (PEG) has a low adverse immune reaction, which is considered to prolong the release time of the complex, and as a filler, it is commonly used in nerve cannulas. Being a hydrogel, it can be affected by polyethylene glycol and change its mechanical properties, so is widely used in the field of cosmetics.

#### 4.2.3. Polycaprolactone (PCL)

Polycaprolactone (PCL) is a kind of aliphatic polyester. There are two main methods for the synthesis of polycaprolactone. One is the polycondensation of hydroxycarboxylic acid; the other is the ring-opening polymerization of lactone [47]. PCL polymer will be hydrolyzed into low molecular weight oligolactone (OCL) under in vivo conditions. Next, macrophages and giant cells eliminate larger fragments by phagocytosis, and finally, esterase degrades smaller fragments in cells [48]. The degradation of PCL is prolonged and takes up to 24 months [49]. Therefore, some studies prefer to classify it as a non-degradable biomaterial.

#### 4.2.4. Composite Material

The hydrophobicity of PLA or the non-adhesion of PCL make more and more scholars devote themselves to the research and development of composite materials. Luis et al. used a neural tube made of DL-lactide-ε-caprolactone copolyester to bridge the sciatic nerve’s 10 mm defect in rats [50]. Polymeric polymers can also be mixed to obtain better properties and higher surface adhesion, such as PCL-PLGA scaffolds. Alternatively, synthetic biomaterial scaffolds are becoming convenient because of their excellent mechanical properties, pure chemical composition, and degradation rate. However, these synthetic scaffolds lack biometric recognition. The major challenge in developing a scaffold, lies primarily in the choice of a blend of biomaterials with the correct combination of properties [51]. If these composites can better combine bioactive factors such as stem cells, they may promote better repair.

## 5. Other Material

Many other materials, such as fiberglass, ceramics, and metals, have been implanted into damaged peripheral nerves to achieve regeneration. For example, some mixtures of metal magnesium are implanted into the nerve conduit, which can satisfy degradability and increase electrical conductivity, in an attempt to achieve a better effect on nerve repair.

## 6. Conduit Spatial Structure and Fillers

Using natural or synthetic polymers as raw materials, various techniques are used to prepare peripheral nerve regeneration scaffolds. If the production process is different, its biological and physicochemical properties are also different. Melt extrusion is one of the earliest processes used in the manufacture of neural scaffolds. However, this process is only suitable for some thermoplastic materials. The high temperature in the production process can lead to denaturation and deactivation of materials, and now that more attention is paid to the biocompatibility of materials, this process has been phased out [52,53]. Another method, is the freeze-drying integrated molding process, in which active materials, such as proteins, or easily degradable materials, such as polysaccharides, can be processed [54]. The products produced in this way have higher porosity and are conducive to cytokine infiltration, vascular component exchange, and nutrient transmission. Other processes such as enzyme digestion, or chemical treatment can meet different needs. Given the discovery of more and more excellent materials and the maturity of technology, 3D printing has gradually become the mainstream mode of catheter production. In this process, ducts can be arbitrarily edited into the structure they need, or a specific pore structure can be mixed with a variety of materials such as living cells [55,56]. In addition, some techniques incorporate nanofibers into printed structures to simulate extracellular matrix to achieve higher biocompatibility.

### 6.1. Electrospun Nanofibrous Yarns and Conduits

Electrospinning is a multi-purpose and cost-effective process for producing non-woven micro/nanofibers, which can provide an excellent ECM-like environment for tissue and cell growth [57,58]. Electrospun fibers can meet various structural requirements for tissue engineering scaffolds, such as porosity, pores that communicate with each other, adjustable void size, functional filaments, and adjustable morphology [59,60,61]. Electrospinning is a standard method for the preparation of micro-nano fiber [62]. The electric field strength can adjust the diameter of the fiber, mainly by adjusting three parameters in the spinning process: the applied voltage, the defined flow rate, and the distance between the needle and the grounding electrode [63]. The material, shape, and the size of the receiving device, can also change the electric field’s strength and shape, affecting the fiber’s diameter; Figure 1A clearly describes this process. Electrospun micro/nanofilaments can produce a high specific surface area, so that the slow release and diffusion of therapeutic drugs can be achieved through the slow degradation of polymers. The therapeutic drug used in electrospinning, when released, reaches the therapeutic concentration locally and is released continuously; avoiding the damage to other organs that can be caused by systematic administration [64,65]. It is reported that the fiber-containing NGF, can be obtained by electrospinning polycaprolactone and ethyl ethylene copolymer directly, mixed with human *β* nerve growth factor (NGF) [66]. Due to phase separation, NGF encapsulated in the electrospinning solution is uniformly distributed in the fiber during aggregation. Bioactive NGF can be released in vitro for at least three months, especially when the spun fiber is made to have a unique three-dimensional configuration, as shown in Figure 1B. Some researchers show that, because of their special potential, blends of PHAs may be used in the manufacture of multichannel and electrospun NGCs [67]. From in vivo studies, it was also found that sustained-release could be achieved. Results of the total number of myelinated axons, a cross-sectional area of a regenerated nerve, and the G ratio, confirmed that the directional electrospinning fiber conduit, encapsulated by glial cell line-derived neurotrophic factor (GDNF), could significantly promote the regeneration of rat sciatic nerve [68]. Bovine serum albumin (BSA) was added to the spinning solution of PCL and NGF for electrospinning. The addition of BSA not only increased the total amount of NGF released from the filaments, but also improved the biological activity of the released NGF and promoted the neurite formation of PC12 cells. As shown in Figure 1C, the nerve sleeve made of this electrospinning structure can be compounded with a variety of nutritional factors to promote the growth of axons. Therefore, electrospinning technology has been successfully applied to the preparation of peripheral nerve injury conduits [69].

### 6.2. Surface Modification Technology

The interaction between cells and carriers is primarily limited by the surface characteristics and structure of the vessel. Although nanofibers can mimic the morphology of extracellular matrix, some modifications are still needed [70,71]. These are necessary to obtain essential characteristics that can respond to specific biological signals and improve cell attachment, cell proliferation, cell differentiation, and tissue regeneration [72,73,74]. The adhesion of cells can be improved by changing the catheter’s inner surface roughness; Figure 2A,B shows the process of this surface modification. For example, García-Gareta et al.’s data showed that CaP deposits would promote initial attachment, proliferation, and osteogenic differentiation of BM-MSCs [75]. The graphene oxide layer can be sprayed on the surface to improve the roughness, wrinkling, hydrophilicity, and electrical conductivity of nanofibers. Studies have shown that electrical stimulation can guide axon orientation and axon extension, thus promoting nerve regeneration, and Ghasemi-Mobarakeh et al. have summarized the different modification methods that can be carried out to modify conductive biomaterials [76]. Therefore, coating or mixing conductive compounds in scaffolds can improve cell activity, neural signal transduction, extensive dendritic branches, axonal growth, nerve proliferation, migration, and differentiation, resulting in functional recovery. Therefore, it may be a promising direction to imitate neural conductivity with biomaterials [77,78,79]. As mentioned above, the use of nerve repair cells, as shown in Figure 2C, and surface-modified nerve conduit materials, can promote cell attachment and proliferation. These can then be made into a tube; refer to the legend for Figure 2D. Surface modification technology is of great significance in polymerizing polymer repair materials. Whether it is through surface spraying to improve the roughness of the casing surface and promote the attachment of growth factors and cells, or through multi-layer three-dimensional structures to increase the specific surface area of the casing and change the electrical conductivity; all provide potential possibilities for the repair of injured nerves.

## 7. Conclusions and Prospects

As peripheral nerve injures caused by various injuries can seriously reduce patients’ quality of life, polymerized nerve stents can be used to promote the injury repair, especially when the defect is large. Therefore, researchers worldwide are working hard to promote efficient medical devices and biomaterials to promote the effectiveness of neural rehabilitation. Some synthetic polymer materials have been widely used in the clinic, with a high efficiency of nerve repair, few side effects, and some characteristics that traditional nerve repair materials do not have. These include an appropriate degradation rate, designable material structure, variable surface characteristics, and variable electrical conductivity. On the other hand, synthetic scaffolds face many challenges because of their high hydrophobicity, low cell adhesion, and potential rejection. The application of electrospinning inner plant and surface modification technology, has dramatically expanded its application to change these defects. In the clinic, in order to improve the repair effect of peripheral nerve injury, polymer stenting is only one of the links, and other techniques, such as finding suitable electrical stimulation, are also essential. In the future, polymer materials with better biocompatibility may have a high prospect in biological tissue engineering for nerve repair.

## Figures and Tables

**Figure 1 molecules-26-02712-f001:**
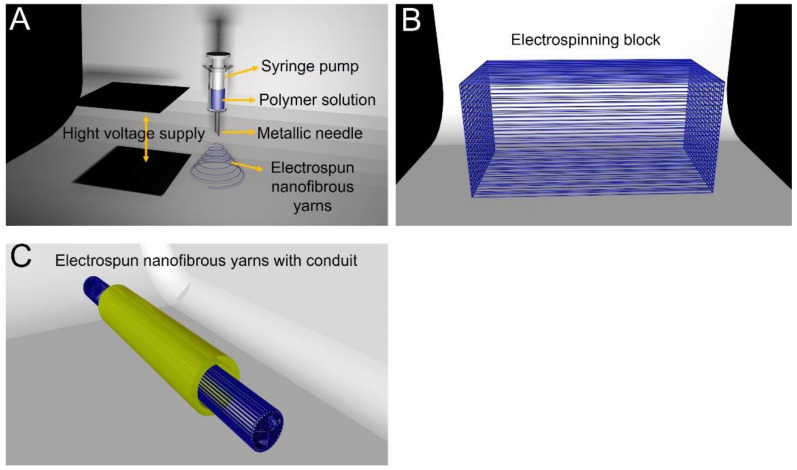
(**A**) Electrospinning fiber production process. (**B**) Electrospinning fiber is made into an electrospinning block with a specific structure. (**C**) The electrospinning block is filled into the nerve conduit to suture the broken end of the peripheral nerve.

**Figure 2 molecules-26-02712-f002:**
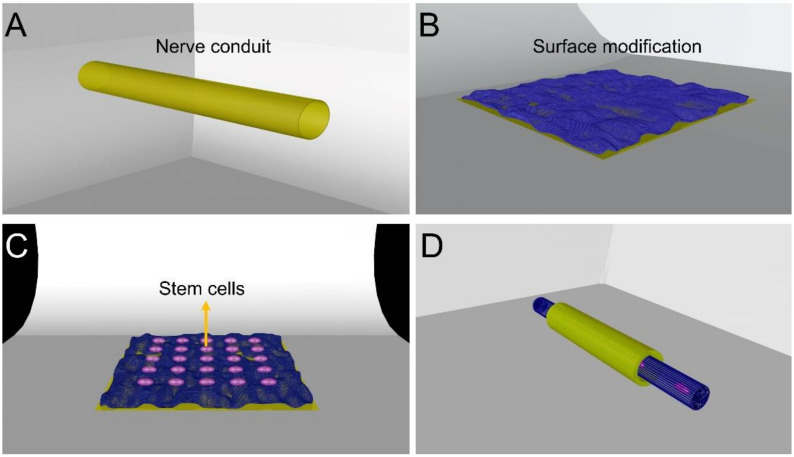
(**A**) Common nerve conduit model surface. (**B**) Use of specific surface modification, to change the roughness of the surface or the three-dimensional surface area. (**C**) This surface-modified nerve conduit material can promote cell attachment and proliferation. (**D**) Nerve catheter made of cells that promote peripheral nerve regeneration such as Schwann cells.

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
