# Peer review of "Polymer Scaffolds for Biomedical Applications in Peripheral Nerve Reconstruction"

_molecules, 2021, doi:10.3390/molecules26092712_

Round 1

Reviewer 1 Report

Comments to Manuscript molecules-1162527

There are some interesting manuscripts that authors did not consider in the review (please, see below):

Journal of Biomedical Science volume 16, Article number: 108 (2009) .

J Tissue Eng Regen Med 2011;5: e17–e35

Biomedical Materials, Volume 12, Number 1 (2017) 015008

Eng. Life Sci. 2015, 15, 612–621

I suggest incorporating the references above mentioned in the adequate sections.

Reviewer 2 Report

This is a comprehensive review of different polymer materials and techniques used in peripheral nerve regeneration. The authors first cover natural materials (chitosan, ECM derivatives such as collagen, laminin and alginate) and then discuss synthetic non-degradable and degradable polymers (PLA, PGA, PEG, PCA). At the end of the review authors touch upon electrospinning and nanofiber use in nerve regeneration.

Overall the review is accurate and broadly samples the field. Use of English is not always correct, which makes it difficult to understand the manuscript at specific points. Please make the following corrections:

  1. Line 28: “damaged by external force hurt New-onset…” I cannot understand what the authors mean here, please rephrase.
  2. Line 37: “new polymer scaffolds nerve conduits…”, please remove the word “scaffolds”.
  3. Line 38: “of polymer scaffolds nerve conduits…”, please remove the word “scaffolds”.
  4. Line 41-47: “Pubmed/Medline…treatment mechanism.”, please remove the entirety of this text, as it is unnecessary. Readers do not need to know which search engines you used.
  5. Line 49: “discusses the possible mechanism, advantages…”, please change to “mechanisms”.
  6. Line 54-55: “cannot be achieved, the peripheral, substantial…”, difficult to understand the authors as this is a grammatically incorrect sentence. Please rephrase.
  7. Line 65: “..full of promoting elements, providing…”, such as? Please provide examples.
  8. Line 77: “However, Physically guided…”, please change to “physically”.
  9. Line 97: “Professor, Saeed Azizi’s application…”, please change to “Author Name et al…. + reference at the end”.
  10. Lines 137-138: “…it is considered a potential material.”, a potential material for what? Please elaborate.
  11. Line 148: “…10mm sciatic nerve defects in rats, and 15mm sciatic nerve defects in rats.”, please combine to one sentence (e.g. “10-15mm nerve defects…”).
  12. Line 160-162: “effectively promotes peripheral nerve regeneration than the silicone conduit”, grammatically incorrect sentence, please rephrase.
  13. Line 190-192: “PLA conduits facilitate…autologous nerve grafts.”, please add “and“ between “injury” and “are considered”.
  14. Line 305-306: “Surface modification technology…polymerizing polymer repair materials”, this sentence does not provide much information and does make much sense. Please elaborate further with 1-2 more sentences on your point here.

Figures 1 and 2 are not of high quality (appear pixelated) and legends hardly provide any information. I also fail to see the point of these figures. Please add:

  1. High quality jpgs of these images
  2. Detailed legends describing the different subpanels (A, B, C) of every figure.
  3. In your manuscript refer to each subpanel individually (e.g. Figure 2B) and use the image to clarify your point. This will also highlight the usefulness of these figures.

Finally, there is a very interesting, recent study on the use of alginate for stem cell differentiation for neural tube modelling and regeneration. Please refer to the following study under section 3.2.3:

Orla M. Fannon, Angela Bithell, Benjamin J Whalley, Evangelos Delivopoulos, “A fibre alginate co-culture platform for the differentiation of mESC and modelling of the neural tube”, Frontiers in Neuroscience, vol. 14, 2020

Reviewer 3 Report

This is an informative and concise review article summarizing materials for peripheral nerve reconstruction that could be of interest to readers. A few suggestions are:

 - Multiple typos regarding wrongly capitalized words and the opposite. 

- Addition of current research is necessary in this review. More than 60% of the references are ten years old or more

- Figures provide little to no information and there is no caption describing the panels.

Round 2

Reviewer 3 Report

The authors have addressed this reviewer's concerns.